# The Positive Effect of Four-Week Combined Aerobic–Resistance Training on Body Composition and Adipokine Levels in Obese Females

**DOI:** 10.3390/sports11040090

**Published:** 2023-04-20

**Authors:** Purwo Sri Rejeki, Adi Pranoto, Ilham Rahmanto, Nabilah Izzatunnisa, Ghana Firsta Yosika, Yetti Hernaningsih, Citrawati Dyah Kencono Wungu, Shariff Halim

**Affiliations:** 1Physiology Division, Department of Medical Physiology and Biochemistry, Faculty of Medicine, Universitas Airlangga, Surabaya 60132, East Java, Indonesia; 2Doctoral Program of Medical Science, Faculty of Medicine, Universitas Airlangga, Surabaya 60132, East Java, Indonesia; 3Medical Program, Faculty of Medicine, Universitas Airlangga, Surabaya 60132, East Java, Indonesia; 4Study Program of Sports Coaching Education, Faculty of Teacher Training and Education Universitas Tanjungpura, Pontianak 78124, West Kalimantan, Indonesia; 5Department of Clinical Pathology, Faculty of Medicine, Universitas Airlangga, Surabaya 60132, East Java, Indonesia; 6Biochemistry Division, Department of Medical Physiology and Biochemistry, Faculty of Medicine, Universitas Airlangga, Surabaya 60132, East Java, Indonesia; 7Clinical Research Centre, Management and Science University, Shah Alam 40100, Selangor, Malaysia

**Keywords:** adipokines, combined training, inflammation, metabolic syndrome, obesity

## Abstract

Obesity is a metabolic disease that is caused by a lack of physical activity and is associated with an increased risk of chronic inflammation. A total of 40 obese adolescent females with an average age of 21.93 ± 1.35 years and average body mass index (BMI) of 30.81 ± 3.54 kg/m^2^ were enrolled in this study, randomized, and divided into four groups, i.e., control (CTL; *n* = 10), moderate intensity aerobic training (MAT; *n* = 10), moderate intensity resistance training (MRT; *n* = 10), and moderate intensity combined aerobic–resistance training (MCT; *n* = 10). The enzyme-linked immunosorbent assay (ELISA) kits method was used to analyze the adiponectin and leptin levels between pre-intervention and post-intervention. Statistical analysis was conducted using a paired sample *t*-test, while correlation analysis between variables used the Pearson product–moment correlation test. Research data showed that MAT, MRT, and MCT significantly increased adiponectin levels and decreased leptin levels compared to the CTL (*p* ≤ 0.05). The results of the correlation analysis of delta (∆) data showed that an increase in adiponectin levels was significantly negatively correlated with a decrease in body weight (BW) (r = −0.671, *p* ≤ 0.001), BMI (r = −0.665, *p* ≤ 0.001), and fat mass (FM) (r = −0.694, *p* ≤ 0.001) and positively correlated with an increase in skeletal muscle mass (SMM) (r = 0.693, *p* ≤ 0.001). Whereas, a decrease in leptin levels was significantly positively correlated with a decrease in BW (r = 0.744, *p* ≤ 0.001), BMI (r = 0.744, *p* ≤ 0.001), and FM (r = 0.718, *p* ≤ 0.001) and negatively correlated with an increase in SMM (r = −0.743, *p* ≤ 0.001). In summary, it can be concluded that our data show that adiponectin levels increased and leptin levels decreased after the intervention of aerobic, resistance, and combined aerobic–resistance training.

## 1. Introduction

The prevalence rate of obesity in the world has continued to increase over the last 50 years and it is therefore considered to be an epidemic or even a pandemic in the 21st century [1,2]. Data reported by the NCD Risk Factor Collaboration (NCD-RisC) state that the population with obesity has nearly tripled from 1975 to 2016 and is dominated by women compared to men [3]. This condition will increase the risk of chronic disease morbidity in an individual, namely presenting with depression, type two diabetes mellitus, cardiovascular disease, and cancer to cause increased mortality [4]. Obesity is also suspected of inducing a chronic inflammatory condition characterized by a higher proportion of M1 (pro-inflammatory) macrophages than M2 (anti-inflammatory) types [5]. M1 macrophage accumulation in adipose tissue will produce various pro-inflammatory cytokines and chemokines that contribute to insulin resistance, such as tumor necrosis factor-alpha (TNF-α), interleukin 6 (IL-6), resistin, and leptin [6,7]. Whereas, anti-inflammatory cytokines produced by M2 macrophages such as adiponectin, interleukin 10 (IL-10), and omentin actually decrease in obese conditions [6].

Recent terminology defines adipose tissue as a complex organ consisting of various types of cells with various functions, such as energy storage, metabolic regulation, neuroendocrine function, and immunity [6]. Adipose tissue can also produce adipocytokines such as leptin and adiponectin which play an important role in controlling metabolic processes synergistically or antagonistically, especially in obesity [8]. Leptin has a major effect of suppressing appetite and stimulating energy expenditure through the Janus kinase signaling cascade involving a signal transducer and activator of transcription (JAK-STAT) [9]. In some individuals with obesity, the brain has difficulty responding to leptin, so plasma leptin concentrations will continue to be produced; this is known as leptin resistance [9,10]. On the other hand, adiponectin, as an anti-inflammatory mediator, has a function in the regulation of glucose and fatty acid metabolism, the inflammatory response, and insulin sensitivity through the activation of the AMP-activated kinase (AMPK) and peroxisome proliferator-activated receptors (PPARs) pathways [8,11]. Obese conditions tend to decrease adiponectin secretion as a result of an increase in pro-inflammatory agents in adipose tissue and are negatively correlated with increased triglycerides, causing insulin resistance [12].

Training that induces weight loss has been shown to improve obesity-related comorbidities such as insulin resistance, inflammation, dyslipidemia, and cardiovascular disease [12,13]. Several studies have reported that different types of training, such as aerobic, resistance, or combination (aerobic and resistance) training, in obese individuals have a positive effect in terms of reducing serum leptin levels and increasing serum adiponectin levels [14,15,16,17]. The training frequency and intensity were also reported to have positive effects in reducing pro-inflammatory biomarkers and increasing anti-inflammatory biomarkers, as well as insulin sensitivity [18]. However, another study reported that adiponectin levels did not change significantly after aerobic and resistance training, and leptin levels were also found not to decrease significantly after 10 weeks of combined training [19,20]. Therefore, the fundamental impact of different types of training on adipokine levels is still unclear. This study aims to prove differences in the effect of the type of training (aerobic, resistance, and combined aerobic–resistance training) in moderate intensity exercise with a frequency of three times per week for a duration of four weeks on the body composition and adipokine levels in obese adolescent females. We assume that all the participants performed the same amount of exercise (12 training sessions) and that the 3 types of exercise have the same energy expenditure although the type of muscle work differs. There are no imposed conditions in this study. It is hoped that the results of this study can guide the application of several types of training in preventing or overcoming the problem of obesity by using body composition and adipokine parameters. The results of this study can also be used as a basis for physical therapy in reducing the impact of morbidity, mortality, and improving the quality of life of obese people.

## 2. Materials and Methods

### 2.1. Research Study Design

This study used the true experimental method with the randomized pretest–posttest control group design. A total of 40 obese adolescent females the following averages voluntarily participated in the study: aged 21.93 ± 1.35 years, body mass index (BMI) 30.81 ± 3.54 kg/m^2^, blood pressure (systolic 113.48 ± 1.44 mmHg and diastolic 80.65 ± 1.17 mmHg), resting heart rate (RHR) 79.45 ± 1.24 bpm, oxygen saturation (SpO_2_) 97.70 ± 1.90%, fasting blood glucose (FBG) 91.70 ± 5.76 mg/dL, and hemoglobin (Hb) 15.38 ± 1.92 g/dL. Before we involved participants in the study, we first screened their level of physical activity using the Global Physical Activity Questionnaire (GPAQ). The average respondent involved in this study has a physically active habit because their GPAQ results achieve a metabolic equivalent task (MET) of 600 min or more per week and at the time of the study onset, we did not recommend changing these habits. The calculation of the study sample size was carried out using the Higgins Kleinbaum formula [21] with reference values from previous similar studies so that a minimum sample size was obtained and 40 participants were taken [22]. All participants filled out and signed an informed consent form after obtaining information about the study.

### 2.2. Aerobic, Resistance, and Combined Aerobic–Resistance Training Intervention

The physical training intervention was implemented and supervised by personal trainers from Atlas Sports Club Malang, East Java, Indonesia, to ensure that the exercises carried out by the subject were correct and reduced the risk of injury. The training program implemented in this study used three types of training, namely, moderate intensity aerobic training (MAT; *n* = 10), moderate intensity resistance training (MRT; *n* = 10), and moderate intensity combined aerobic–resistance training (MCT; *n* = 10). In addition, this study used a control group (CTL; *n* = 10) to be compared with the intervention group. The control group did not receive any training program in this study and was instructed not to change extreme activity habits or dietary habits. Subjects who met the inclusion criteria which included female sex; aged 20–24 years; BMI 27.5–37.5 kg/m^2^; fat mass ≥ 30%; and were in good health as indicated by normal blood pressure, resting heart rate, oxygen saturation, fasting blood glucose, and hemoglobin would be randomized and divided into four groups (CTL, MAT, MRT, and MCT) equally. Aerobic training is carried out in 135 min per week by running on a treadmill with an intensity of 60–70% HRmax which is carried out for 45 min each session (5 min of warm-up, 35 min of core training, and 5 min of cool-down). Resistance training is carried out with an intensity of 60–70% 1-RM which is carried out for 6 sets of 15 reps with active rest between sets of 30 s. The method used in resistance training is circuit training and is divided into upper body and lower body. Upper body resistance training includes pull-down, shoulder press, chest press, and tricep push-down, while lower body resistance training includes leg press, leg extension, leg curls, and barbell squat press. Combined aerobic–resistance training is carried out by combining aerobic training with resistance training which is carried out on different days (lower body resistance training, upper body resistance training, and aerobic training). Warming up and cooling down for the three types of training were each carried out for 5 min with an intensity of 50% HRmax which was performed by walking on a treadmill. Training was carried out with a frequency of three times per week (Monday, Wednesday, and Friday) for four weeks. In this study, several important factors such as the intensity, volume, and frequency of exercise were equated, while the difference was only the type of exercise (aerobic, resistance, and combined aerobic–resistance training). Both MTT, MRT, and MCT exercise sessions were matched for caloric expenditure (400 kcal). Participants arrived at the Atlas Sports Club Malang at approximately 07.00 a.m. Participants’ heart rate during training were monitored using a polar h10 heart rate monitor. The study environment used had a humidity level of 50–70% with a room temperature of 26 ± 1 °C [23,24]. In this study, there was no specific monitoring of the subjects’ diets. We instructed all subjects to only adjust their diet to ensure their last food intake was at 9 p.m. each time they were going to carry out an exercise program (Monday, Wednesday, and Friday) and they were not allowed to make significant changes to their previous diet. As long as the diet was set, the subject was still allowed to drink something that does not contain calories to maintain hydration.

### 2.3. Blood Sample Collection and Adipokine Level Analysis

As much as 4 mL of blood was taken from the cubital vein after fasting overnight for 12 h [25]. Blood sampling was carried out twice, namely 30 min pre-intervention (0 weeks) and 24 h post-intervention (4 weeks) to ensure that the blood samples taken were the result of chronic physical training, and not acutely. The collected blood was then centrifuged for 15 min at 3000 rpm [25]. Then, the serum was separated and immediately analyzed for adiponectin levels using the ELISA Kit (REF.CAN-APN-5000; Diagnostics Biochem Canada (DBC), Inc. London, ON, Canada) and the analysis of the leptin levels used the ELISA Kit (REF.CAN-L-4260; Diagnostics Biochem Canada (DBC), Inc. London, ON, Canada). The body height was measured using a portable stadiometer seca 213 (Seca-213, Hammer Steindamm 3–25 22,089 Hamburg Germany). The body weight, body mass index, fat mass, free fat mass, fat mass index, and free fat mass index of skeletal muscle were measured using a Seca mBCA 554 (Seca mBCA 554 Medical Body Composition Analyzer, Hammer Steindamm 3–25 22,089 Hamburg Germany). The blood pressure and resting heart rate were measured using the Omron Digital Ten-simeter—HBP 9030 tool at the non-dominant arm. The oxygen saturation was measured using the Beurer Pulse Oximeter PO 30 tool. The body temperature was measured using the Omron Model MC-343F tool. The fasting blood glucose and hemoglobin levels were measured using the Accu Chek Performa and Easy Touch GCHb tools at the middle fingertip.

### 2.4. Statistical Analysis

Data analysis conducted in this study used several stages, namely descriptive tests, normality tests, and homogeneity using the Shapiro–Wilk test and Levene test. If the data were normally distributed and had a homogeneous variant, a parametric Paired Samples T-Test, One-way ANOVA, and a follow-up test with Tukey’s HSD posthoc test were performed. However, for data that were not normally distributed and did not have a homogeneous variant, the non-parametric Kruskal–Wallis test was performed and the Mann–Whitney U test was continued. The correlation between variables used Pearson’s product–moment correlation test. All data are displayed with mean ± SD. All statistical analyses used a significant level of 5%.

## 3. Results

The analysis of the basic data on the characteristics of the research subjects showed that there were no significant differences in the four groups (*p* ≥ 0.05) (Table 1). Therefore, the four groups are at the same starting point so that if there is a change in body composition and adipokine levels it is most likely due to the effects of training intervention. The results of the analysis of body composition and adipokine levels pre-intervention and post-intervention in each group are presented in Figure 1, Figure 2, Figure 3, Figure 4, Figure 5 and Figure 6 below.

Based on Figure 1, it can be seen that there was an increase in the mean body weight in CTL between pre-intervention and post-intervention, whereas in MAT, MRT, and MCT it showed a decrease. The analysis of the mean body weight pre-intervention and post-intervention on CTL (75.97 ± 10.61 kg vs. 78.42 ± 10.42 kg, *p* = 0.000), MAT (74.88 ± 7.45 kg vs. 73.33 ± 8.25 kg, *p* = 0.005), MRT (75.28 ± 9.85 kg vs. 73.14 ± 9.82 kg, *p* = 0.000), and MCT (75.02 ± 11.65 kg vs. 71.82 ± 11.39 kg, *p* = 0.000).

Based on Figure 2, it can be seen that there is an increase in the mean body mass index in CTL between pre-intervention and post-intervention, whereas in MAT, MRT, and MCT it shows a decrease. The analysis of the mean body mass index pre-intervention and post-intervention on CTL (30.72 ± 3.75 kg/m^2^ vs. 31.72 ± 3.71 kg/m^2^, *p* = 0.000), MAT (30.82 ± 3.55 kg/m^2^ vs. 30.16 ± 3.65 kg/m^2^, *p* = 0.010), MRT (30.96 ± 3.88 kg/m^2^ vs. 30.08 ± 3.86 kg/m^2^, *p* = 0.000), and MCT (30.76 ± 3.53 kg/m^2^ vs. 29.44 ± 3.50 kg/m^2^, *p* = 0.000).

Based on Figure 3, it can be seen that there is an increase in the mean fat mass in CTL between pre-intervention and post-intervention, whereas in MAT, MRT, and MCT it shows a decrease. The analysis of the mean fat mass pre-intervention and post-intervention on CTL (34.18 ± 6.94 kg vs. 37.43 ± 5.66 kg, *p* = 0.000), MAT (34.06 ± 4.99 kg vs. 32.96 ± 5.12 kg, *p* = 0.000), MRT (33.96 ± 6.61 kg vs. 31.87 ± 6.56 kg, *p* = 0.000), and MCT (33.72 ± 4.89 kg vs. 30.22 ± 4.20 kg, *p* = 0.000).

Based on Figure 4, it can be seen that there is a decrease in the mean skeletal muscle mass in CTL between pre-intervention and post-intervention, while in MAT, MRT, and MCT it shows an increase. The analysis of the mean skeletal muscle mass pre-intervention and post-intervention on CTL (19.48 ± 2.76 kg vs. 17.61 ± 2.67 kg, *p* = 0.000), MAT (18.69 ± 2.08 kg vs. 19.99 ± 2.21 kg, *p* = 0.000), MRT (19.71 ± 4.48 kg vs. 21.92 ± 4.45 kg, *p* = 0.000), and MCT (20.02 ± 3.17 kg vs. 23.53 ± 3.58 kg, *p* = 0.000).

Based on Figure 5, it can be seen that the mean adiponectin levels in CTL between pre-intervention and post-intervention tended to be the same, whereas in MAT, MRT, and MCT showed an increase. The analysis of the mean adiponectin levels pre-intervention and post-intervention of CTL (10.69 ± 4.18 ng/mL vs. 10.32 ± 2.59 ng/mL, *p* = 0.700), whereas, MAT showed a significant difference (10.50 ± 2.52 ng/mL vs. 16.59 ± 3.72 ng/mL, *p* = 0.001) as did MRT (10.79 ± 2.05 ng/mL vs. 18.06 ± 5.73 ng/mL, *p* = 0.002) and MCT (10.73 ± 2.16 ng/mL vs. 24.35 ± 6.93 ng/mL, *p* = 0.000).

Based on Figure 6, it can be seen that the mean leptin levels in the CTL between pre-intervention and post-intervention tend to be the same, whereas in MAT, MRT, and MCT it shows a decrease. The analysis of the mean leptin levels pre-intervention and post-intervention of CTL (25.34 ± 5.90 ng/mL vs. 25.80 ± 6.49 ng/mL, *p* = 0.345), whereas MAT showed a significant difference (24.99 ± 7.18 ng/mL vs. 20.64 ± 4.03 ng/mL, *p* = 0.004) as did MRT (24.80 ± 4.94 ng/mL vs. 19.92 ± 5.48 ng/mL, *p* = 0.000) and MCT (25.14 ± 7.19 ng/mL vs. 13.32 ± 2.10 ng/mL, *p* = 0.000). The results of the analysis of body composition and adipokine levels between groups are presented in Table 2, while the results of the analysis of the correlation between body composition and adipokine levels are shown in Table 3.

## 4. Discussion

This study aims to measure changes in body composition and adipokine levels in 40 obese adolescent females with an average age of 21.93 ± 1.35 years who were divided equally into four groups: CTL, MAT, MRT, and MCT. The exercise intervention was carried out with moderate intensity and at a frequency of three times per week for a duration of four weeks. In the early stages before the intervention, there was no significant or insignificant difference in the distribution of the data, so all participants showed the same initial state in each intervention group and met the established criteria. The main results of this study showed improvements in body composition and adipokine levels in the MAT, MRT, and MCT groups compared to CTL.

Based on data from WHO [26], the prevalence of obesity is higher in women than men due to differences in hormones, so women have a 10% greater proportion of fat than men [27]. Subjects in this study were categorized as obese according to the criteria that have been validated for the Asia–Pacific population, especially Indonesia, namely BMI ≥ 25 [28]. The results of this study showed significant improvements in body composition (BW, BMI, FM, and SMM) and adipokine levels (adiponectin and leptin) in the MAT, MRT, and MCT groups compared to CTL (Figure 1, Figure 2, Figure 3, Figure 4, Figure 5 and Figure 6). This finding was dominated by the MCT group which was shown with the highest Δ Post-test–Pre-test value in Table 2. The results of univariate correlations analysis on the Δ parameter values tested against the Δ values of adiponectin and leptin showed a significant correlation (*p* ≤ 0.001), as shown in Table 3.

The best improvement in body composition and adipokine levels occurred in the MCT group, followed by the MRT group, and then the MAT. The differences in the mechanism of action between aerobic and resistance-type training on the parameters BW, BMI, FM, and SMM are the background to the differences in the results of serum adiponectin and leptin levels. Training modulates white adipose tissue (WAT) browning so that it induces fat loss and weight loss [29]. Reducing the training-induced excess accumulation of adipose tissue will decrease anti-inflammatory adipocytokines (adiponectin) and increase pro-inflammatory adipocytokines (leptin) [30]. A systematic review and meta-analysis study show that moderate intensity aerobic training for 6–12 months can reduce body weight and waist circumference in both overweight and obese populations [31]. On the other hand, studies have found that regular resistance training changes body composition through muscle hypertrophy which suppresses the proportion of fat mass [16,32].

The results of this study indicate an increase in serum adiponectin levels from highest to lowest in the MCT, MRT, and MAT groups, respectively, followed by a decrease in serum leptin levels successively between groups. In addition, the results of this study also revealed a decrease in serum leptin levels from highest to lowest in the MCT, MRT, and groups, respectively. All groups showed significant differences compared to the control group, but only the MCT group had significant differences from the MRT and MAT groups (Table 2). This study proves that aerobic, resistance, and combined types of training are positively correlated with increased serum adiponectin levels and decreased serum leptin levels (Table 3). These results are consistent with several research studies which also state that these three types of training can significantly improve serum adiponectin and leptin levels in obese patients [14,15,16,33,34,35,36].

Aerobic training induces FA metabolism through oxidation in muscle mitochondria to produce large amounts of ATP [37]. Conditions in aerobic training that require high ATP will trigger the activation of the AMPK pathway which acts to conserve ATP by inhibiting biosynthetic and anabolic pathways (glycogen and protein synthesis), as well as stimulating catabolic pathways by increasing glucose transport and fat metabolism [38]. Meanwhile, resistance training that focuses on muscle hypertrophy will induce the release of anti-inflammatory myokines and activate the AMPK and phosphatidylinositol 3-kinase (PI3-kinase) pathways resulting in muscle–adipose crosstalk which underlies fat-burning in the body [39,40,41]. Combination training uses aerobic and resistance mechanisms of action to improve adipocytokine profiles. Resistance training focuses on the use and hypertrophy of a few muscle groups in the extremities, whereas aerobic training induces almost all body muscle tension [39]. In addition, aerobic training is often associated with greater energy expenditure, whereas resistance training is more associated with maintaining muscle mass which shifts fat mass composition [39]. It can be concluded that optimal fat burning in combination training underlies an increase in serum adiponectin levels and a decrease in serum leptin levels in the body.

Adiponectin is responsible for controlling all energy in the body, the inflammatory response, insulin sensitivity, and fat oxidation processes by activating the AMPK and PPAR pathways [11]. Adiponectin in carrying out its role will bind to its two main receptors, namely Adiponectin Receptor 1 (AdipoR1) and Adiponectin Receptor 2 (AdipoR2) [42]. AdipoR1, which is abundant in skeletal muscle, can activate AMPK to induce muscle fat oxidation and glucose transport in the muscles. Meanwhile, AdipoR2 activates the PPARs pathway which increases the burning of fatty acids and energy consumption [42]. In obese conditions, there is a decrease in adiponectin secretion due to excess accumulation of free fatty acids (FFAs) stored in WAT in the form of triglycerides (TG) [43]. Hypoadiponectinemia in obesity is caused by adipocyte hypoxia, oxidative stress, insulin resistance, and increased pro-inflammatory cytokines such as TNF-α and IL-6 [29]. Training, which modulates fat metabolism, plays a role in maintaining the long-term anti-inflammatory microenvironment and helps restore the WAT inflammatory disorder that occurs in obesity [29]. In other words, training-induced reduction in body weight and body fat mass is the key to returning adiponectin levels to normal.

Leptin is an adipocytokine produced by WAT and has a role in regulating the homeostasis of energy balance and metabolism by increasing energy expenditure and suppressing appetite [44]. In carrying out its duties, leptin will bind to the leptin receptor (LepR) in the brain which facilitates leptin’s pleiotropic effect and activates a negative feedback mechanism between the hypothalamus and adipose tissue to regulate body mass and satiety responses [45]. Leptin can activate the AMPK pathway and inhibit acetyl-CoA carboxylase, thus inducing fatty acid oxidation in skeletal muscle [46]. However, in obese conditions, the phenomenon of “leptin-induced leptin resistance” occurs which is characterized by hyperleptinemia and decreased sensitivity and failure of the brain’s response to leptin, thus inducing an increase in appetite and body weight [44,47]. Studies have found that chronic training can improve the anti-inflammatory environment and reduce oxidative stress in the hypothalamus, thus inducing increased sensitivity and decreased leptin production in obese patients [29]. In conclusion, training is able to re-optimize the function of leptin by improving body composition (decreasing fat mass and browning adipose tissue) and regulating leptin signaling factors [46].

The results of this study indicate an improvement in adipokine levels followed by an improvement in body composition. Basically, both increased serum adiponectin levels and decreased serum leptin are mediated by a decrease in adipose tissue which is characterized by a decrease in body weight, body mass index, fat mass, and free fat mass, followed by an increase in skeletal muscle mass. Optimal fat burning underlies the improvement of adipokine levels in the body [39]. This explains that training can be an effective approach to obesity management. The best improvement in body composition and adipokine levels was obtained from combination training. The maximal effect induced by combination training is a combination framework of aerobic training (improvement of energy systems through oxidative metabolism, qualitative changes in skeletal muscle fiber, metabolic capacity, and cardiorespiratory fitness) and resistance training (increases in strength, fiber diameter, and muscle mass) [48]. On the other hand, combination training also maximizes muscle impact on the upper and lower extremities (resistance) and the whole body (aerobics) [39]. The selection of training intensity has a role in determining the main energy source. The source of energy used in physical training at moderate intensity, especially aerobics, is more dominant from the oxidation of FFAs, while at heavy intensity it will switch to using oxidation from carbohydrates [38]. Therefore, the results of this study can be the basis for recommending the selection of the best type of physical training as a treatment for obesity through a training-based non-pharmacological approach.

This study compared improvements in body composition and adipokine levels using different types of training. However, other factors may have contributed to this result such as nutritional factors and diet monitoring [49]. Consuming certain foods that contain vitamin D, polyphenols, carotenoids, and omega-3 fatty acids has an effect on improving adipokine levels in both animal and human studies [50]. However, these factors were paid less attention in this study and focused more on selecting the type of training, thus not maximizing the possibility of other triggers also initiating these results. Diet monitoring needs to be carried out in future studies so that all participants have the same approach conditions and minimize bias intervention in research results due to diets that are not monitored. Therefore, similar studies using a male population, a larger population size, and enforcement of caloric penalties need to be carried out to get better results.

## 5. Conclusions

In general, it can be concluded that moderate intensity combination training performed three times per week for four weeks is more effective in improving body composition and adipokine levels than aerobic and resistance training. We also found a strong relationship between body composition and adipokine levels in obese women. This study supports previous research evidence which states that aerobic, resistance, and aerobic–resistance training have a positive effect on improving body composition and adipokine parameters. Moderate physical training carried out at a frequency of three times per week for four weeks is sufficient to provide significant and improved results in individuals with obesity. These results can be used as a guide in preventing or overcoming the problem of obesity by using body composition and adipokine parameters while taking into account important factors such as the type of exercise, intensity, and frequency of exercise. The results of this study recommend training, especially combination training, as a treatment for obesity.

## Figures and Tables

**Figure 1 sports-11-00090-f001:**
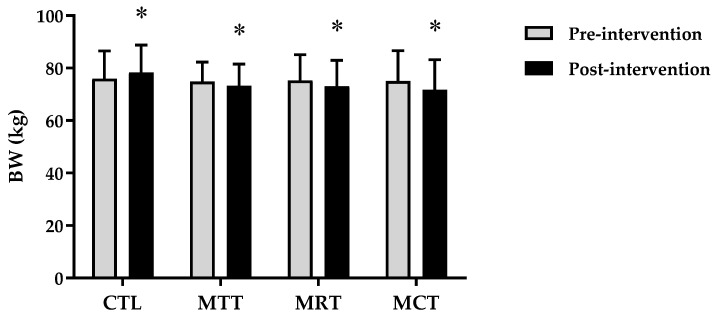
The analysis of BW (kg) between pre-intervention vs. post-intervention in four groups. (*) Significant vs. pre-intervention (*p* ≤ 0.01).

**Figure 2 sports-11-00090-f002:**
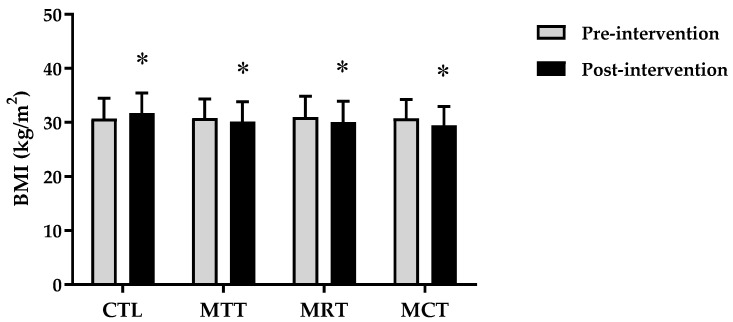
The analysis of BMI (kg/m^2^) between pre-intervention vs. post-intervention in four groups. (*) Significant vs. pre-intervention (*p* ≤ 0.01).

**Figure 3 sports-11-00090-f003:**
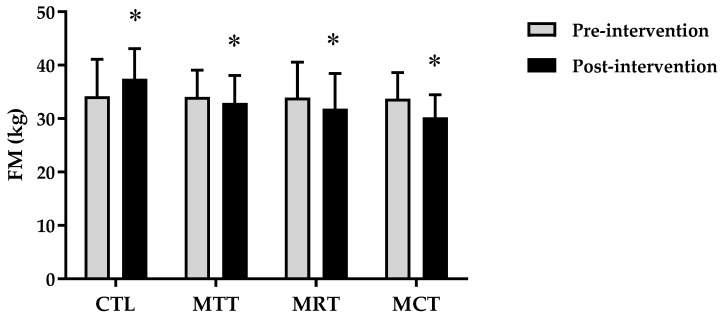
The analysis of FM (kg) between pre-intervention vs. post-intervention in four groups. (*) Significant vs. pre-intervention (*p* ≤ 0.01).

**Figure 4 sports-11-00090-f004:**
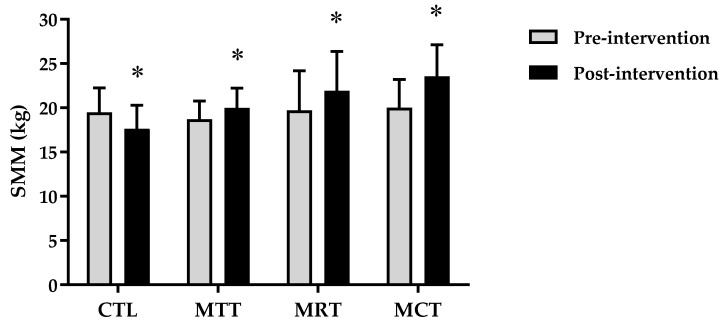
The analysis of SMM (kg) between pre-intervention vs. post-intervention in four groups. (*) Significant vs. pre-intervention (*p* ≤ 0.01).

**Figure 5 sports-11-00090-f005:**
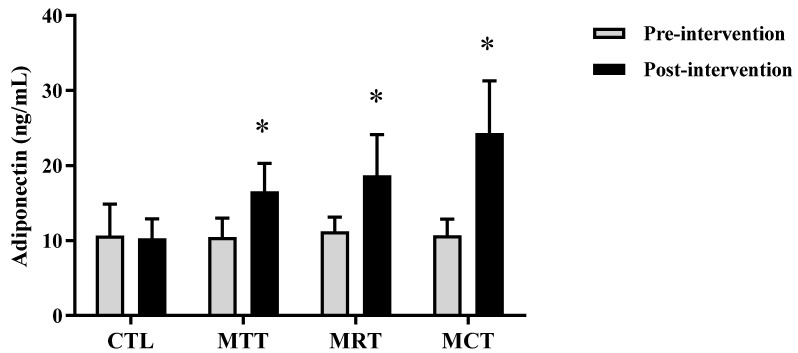
The analysis of adiponectin levels (ng/mL) pre-intervention vs. post-intervention in four groups. (*) Significant vs. pre-intervention (*p* ≤ 0.01).

**Figure 6 sports-11-00090-f006:**
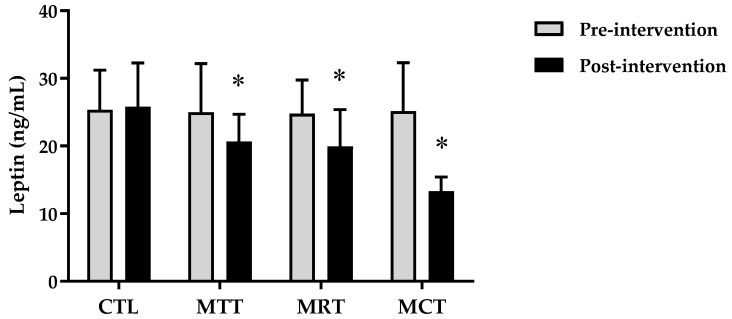
The analysis of leptin levels (ng/mL) pre-intervention vs. post-intervention in four groups. (*) Significant vs. pre-intervention (*p* ≤ 0.01).

**Table 1 sports-11-00090-t001:** General data on the characteristics of the subjects in the four study groups.

Parameters	CTL (*n* = 10)	MAT (*n* = 10)	MRT (*n* = 10)	MCT (*n* = 10)	*p*-Value
Age (yrs)	22.30 ± 1.57	21.60 ± 1.65	21.80 ± 1.23	22.00 ± 0.94	0.704
BW (kg)	75.97 ± 10.61	74.88 ± 7.45	75.28 ± 9.85	75.02 ± 11.65	0.995
BH (m)	1.57 ± 0.06	1.56 ± 0.07	1.56 ± 0.06	1.56 ± 0.05	0.960
BMI (kg/m^2^)	30.72 ± 3.75	30.82 ± 3.55	30.96 ± 3.88	30.76 ± 3.53	0.999
FM (kg)	34.18 ± 6.94	34.06 ± 4.99	33.96 ± 6.61	32.02 ± 6.73	0.854
FM (%)	39.73 ± 7.07	42.71 ± 3.71	44.51 ± 4.17	42.20 ± 3.54	0.194
FMI (kg/m^2^)	12.82 ± 2.71	12.97 ± 2.48	13.85 ± 2.96	13.07 ± 2.29	0.819
FFM (kg)	43.62 ± 5.07	41.92 ± 3.38	41.67 ± 4.51	43.40 ± 5.54	0.718
FFMI (kg/m^2^)	17.50 ± 1.59	17.15 ± 1.35	16.97 ± 1.06	17.77 ± 1.65	0.603
SMM (kg)	19.48 ± 2.76	18.69 ± 2.08	19.71 ± 4.48	20.02 ± 3.17	0.820
WC (cm)	0.92 ± 0.09	0.91 ± 0.10	0.90 ± 0.05	0.89 ± 0.05	0.843
HC (cm)	1.10 ± 0.09	1.11 ± 0.15	1.09 ± 0.05	1.12 ± 0.18	0.944
WHR	0.84 ± 0.05	0.83 ± 0.08	0.83 ± 0.05	0.80 ± 0.09	0.739
SBP (mmHg)	114.80 ± 8.11	110.00 ± 8.21	114.70 ± 10.20	114.40 ± 10.20	0.599
DBP (mmHg)	79.30 ± 8.29	79.10 ± 7.19	82.90 ± 8.31	81.30 ± 5.96	0.635
RHR (bpm)	77.10 ± 9.04	76.60 ± 6.19	82.10 ± 8.50	82.00 ± 6.48	0.219
SpO_2_ (%)	98.00 ± 0.94	96.80 ± 3.22	98.20 ± 1.03	97.80 ± 1.40	0.370
BT (°C)	36.19 ± 0.35	36.25 ± 0.21	36.25 ± 0.23	36.14 ± 0.17	0.711
FBG (mg/dL)	91.00 ± 6.16	92.00 ± 6.04	90.40 ± 6.28	93.40 ± 4.93	0.686
Hb (g/dL)	14.49 ± 2.69	15.69 ± 1.43	15.48 ± 1.29	15.87 ± 1.93	0.390

BH: Body height; BMI: Body mass index; BT: Body temperature; BW: Body weight; DBP: Diastolic blood pressure; FBG: Fasting blood glucose; FFM: Free fat mass; FM: Fat mass; FMI: Fat mass index; FFMI: Free fat mass index; Hb: Hemoglobin; HC: Hip circumference; RHR: Resting heart rate; SBP: Systolic blood pressure; SMM: Skeletal muscle mass; SpO_2_: Oxygen saturation; WC: Waist circumference; WHR: Waist to hip ratio. One-way ANOVA was used to compare the differences among groups. Data are presented as mean ± SD.

**Table 2 sports-11-00090-t002:** The analysis of body composition and adipokine levels between groups.

Time	Group				*p*-Value
	CTL (*n* = 10)	MAT (*n* = 10)	MRT (*n* = 10)	MCT (*n* = 10)	
**BW (kg)**					
Pre-intervention	75.97 ± 10.61	74.88 ± 7.45	75.28 ± 9.85	75.02 ± 11.65	0.995 ^$^
Post-intervention	78.42 ± 10.42	73.33 ± 8.25	73.14 ± 9.82	71.82 ± 11.39	0.482 ^$^
Δ Post–Pre	2.45 ± 0.34	−1.55 ± 1.32 **	−2.14 ± 0.15 **	−3.20 ± 0.48 **†#	0.000 ^$^
**BMI (kg/m^2^)**					
Pre-intervention	30.72 ± 3.75	30.82 ± 3.55	30.96 ± 3.88	30.76 ± 3.53	0.999 ^$^
Post-intervention	31.72 ± 3.71	30.16 ± 3.65	30.08 ± 3.86	29.44 ± 3.50	0.563 ^$^
Δ Post–Pre	0.99 ± 0.19	−0.66 ± 0.64 **	−0.88 ± 0.07 **	−1.31 ± 0.16 **†#	0.000 ^$^
**FM (kg)**					
Pre-intervention	34.18 ± 6.94	34.06 ± 4.99	33.96 ± 6.61	33.72 ± 4.89	0.998 ^$^
Post-intervention	37.43 ± 5.66	32.96 ± 5.12	31.87 ± 6.56	30.22 ± 4.20 *	0.035 ^$^
Δ Post–Pre	3.25 ± 1.42	−1.11 ± 0.18 **	−2.09 ± 0.17 **	−3.50 ± 0.97 **†#	0.000 ^$^
**SMM (kg)**					
Pre-intervention	19.48 ± 2.76	18.69 ± 2.08	19.71 ± 4.48	20.02 ± 3.17	0.820 ^$^
Post-intervention	17.61 ± 2.67	19.99 ± 2.21	21.92 ± 4.45 *	23.53 ± 3.58 *	0.002 ^$^
Δ Post–Pre	−1.87 ± 0.67	1.31 ± 0.29 **	2.21 ± 0.21 **	3.51 ± 0.63 **†#	0.000 ^$^
**Adiponectine (ng/mL)**					
Pre-intervention	10.69 ± 4.18	10.50 ± 2.52	10.79 ± 2.05	10.73 ± 2.16	0.996 ^$^
Post-intervention	10.32 ± 2.59	16.59 ± 3.72 **	18.06 ± 5.73 **	24.35 ± 6.93 **††#	0.000 ^
Δ Post–Pre	−0.37 ± 3.00	6.09 ± 3.85 *	7.28 ± 5.69 *	13.61 ± 6.12 **††#	0.000 ^$^
**Leptin (ng/mL)**					
Pre-intervention	25.34 ± 5.90	24.99 ± 7.18	24.80 ± 4.94	25.14 ± 7.19	0.998 ^$^
Post-intervention	25.80 ± 6.49	20.64 ± 4.03 *	19.92 ± 5.48 *	13.32 ± 2.10 **††##	0.000 ^
Δ Post–Pre	0.47 ± 1.48	−4.35 ± 3.62 **	−4.89 ± 1.74 **	−11.81 ± 5.89 **††##	0.000 ^

^$^ One-way ANOVA, followed by Tukey’s HSD post hoc test, was used to compare the differences among groups. ^ Kruskal Wallis Test, followed by Mann-Whitney U Test, was used to compare the differences among groups. Data are presented as Mean ± SD. * Significant vs. control group (CTL) (*p* ≤ 0.05). ** Significant vs. control group (CTL) (*p* ≤ 0.001). † Significant vs. moderate-intensity treadmill training group (MAT) (*p* ≤ 0.05). †† Significant vs. moderate-intensity treadmill training group (MAT) (*p* ≤ 0.001). # Significant vs. moderate-intensity resistance training group (MRT) (*p* ≤ 0.05). ## Significant vs. moderate intensity resistance training group (MRT) (*p* ≤ 0.001).

**Table 3 sports-11-00090-t003:** Univariate correlations with (Δ) adipokine levels in all participants.

Parameters	(Δ) Adipokine Levels			
	(Δ) Adiponectine (ng/mL)		(Δ) Leptin (ng/mL)	
	*r*	*p*-Values	*r*	*p*-Values
Δ BW (kg)	−0.671 **	*p* ≤ 0.001	0.744 **	*p* ≤ 0.001
Δ BMI (kg/m^2^)	−0.665 **	*p* ≤ 0.001	0.744 **	*p* ≤ 0.001
Δ FM (kg)	−0.694 **	*p* ≤ 0.001	0.718 **	*p* ≤ 0.001
Δ SMM (kg)	0.693 **	*p* ≤ 0.001	−0.743 **	*p* ≤ 0.001
Δ Leptin (ng/mL)	−0.594 **	*p* ≤ 0.001	–	–
Δ Adiponectine (ng/mL)	–	–	−0.594 **	*p* ≤ 0.001

** Significant with *p* ≤ 0.001. (–) No analysis.

## Data Availability

Not applicable.

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
