# Peer review of "The Positive Effect of Four-Week Combined Aerobic–Resistance Training on Body Composition and Adipokine Levels in Obese Females"

_sports, 2023, doi:10.3390/sports11040090_

Round 1

Reviewer 1 Report

Dear authors,

I would like to express my gratitude regarding the opportunity to review this manuscript.

At this stage the manuscript requires several improvements. Below suggestions with line indication:

Line 27 - improve text is incomprehensible.

Line 30 - 35  - the symbol ∆ is used excessively.

Line 80 - the use of different trainings should be subordinated to some criteria, e.g. the assumption should assume that all participants perform the same amount of exercise, have the same energy expenditure, but the type of muscle work differs. There are no imposed conditions in your work. How are you going to justify the advantage of some training if you don't describe why it should bring benefits in a quantifiable way? The study aim should be more clearly presented in the end of the introduction section.

Line 98 - The most effective exercise in the prevention and treatment of obesity should take into account several important factors such as intensity, volume, frequency and type of exercise in the work being assessed lack of this information. Please complete them.

How was the diet of the test subjects monitored?

Line 129 -132  - how were these indicators evaluated? The methodology should be more clearly described. 

Line 163 - 165 - Were differences of up to 3% significant at p=0.001. This with 10 people is unbelievable. Can you share the results of this stats?

Line 169 - 174 - same request as above.

Line 179 - explain this being an increase in the CTL group and a decrease in the MTT, MRT, MCT for FM. How was the diet of the subjects monitored?

Line 188 - the same conclusion as above, only with the opposite effect. How was the diet of the subjects monitored?

Line 116 - do an analysis for repeated results. Compare Pre to Post results. You use both parametric and non-parametric tests at the same time - explain that?

Line 224 - increses were used for correlation - why?

Line 338 - conclusions are not revealing against the background of existing knowledge and literature. What did this work add to existing knowledge?

Discussion section should present paragraphs to improve reading conditions. It should start with the objective of the study, followed by the main findings, and afterward relating these with the literature. It is also important to develop limitations and suggestions for future research at the end of this section.

Author Response

Dear Reviewer. Thank you very much for your kind consideration, your valuable comments, and your comprehensive suggestions for our manuscripts. The constructive suggestions for this manuscript are crucial to improve the betterment of understanding global academic audiences. According to your positive comments, the brief explanation for your valuable feedback is as follows.

Reviewer 2 Report

·         The  article seems to be fine and original.

·         Page 2, Line 91: Please describe how the sample size is calculated?

·         What was the criteria to classify obese females into the 3 groups?

·  Page 3, lines 98 -119: Could you explain more about training programme design for control group.

·       Page 3, Line 105: Please clarify the point you are attempting to make. "core training"

·         Page 3, lines 121-132: Provide a reference for this statement.

·         Page 3, lines 123-124: "namely 30 minutes pre-intervention" and "24 hours post-intervention" On what basis was this determined?

·    The idea behind this work is good but methods need precision and clarity. Could you elaborate more about some points which should be taken into account during the design of the training programme like Individual differences between participants, Nutrition, traditional forms of training, and avoiding injuries.

Author Response

(The authors gave the same response as above.)

Reviewer 3 Report

Abstract

Line 22: “and the risk of increases inflammation”. I am not sure what you mean here. You need to re-write this part in order to make sense as you are explaining the cause of obesity in this sentence.

Line 27: “Statistical analysis using Paired Samples t-Test with a significant level of 5%”. You need to correct the writing here. Maybe statistical analysis was conducted using a paired sample t-test…

Line 29: you introduced a control group here but you haven’t mentioned anything about a control group before. Also, earlier when you presented the 3 types of exercise regimens you should have included in a parenthesis how many participants you had in each exercise group.

Line 30: You present the results of correlations but before you stated that you conducted a paired sample t-test. If you conducted correlations as well you should have included that in the statistical analysis.

In general, in the abstract, you used many abbreviations which makes it difficult for the reader to understand what you found. I understand that there is a limit to the number of words, but the abstract should clearly present what you did and your general findings for someone to decide whether he wants to read the full manuscript or not.

Introduction

Line  42: “so that it can be said to be an”. You can say “therefore it is considered to be” ….

Lines 45-46: “chronic disease morbidity in an individual such as disability …..”. You need to correct this statement. You have included a review study (4) to support this statement which is not really accurate based on the review of Hruby et al (2015).

Good presentation of leptin and adiponectin!

Line 77: in their study as it is not just his study (study number 20)

Methodology

I am not sure that 4 weeks are enough to see the reported changes. What were the activity levels of your participants before they participated in your program? Previous activity levels play an important role in the extent of the differences (changes) that we see when those exercise programs are applied. You need to report that in the description of your participants.

Statistical analysis: you need to consider using the same tense throughout the study. Use past tense as the study and the statistical analysis are already completed.

Line 140:  what do you mean by the Mann-Whitney U test is continued? In the case of non-parametric tests?

Lines 140-141: correct the writing here.

The results section is clearly presented. This is a very good section with excellent tables and figures.

Discussion

Nicely presented and supported.

Author Response

(The authors gave the same response as above.)

Round 2

Reviewer 1 Report

Corrections are sufficient to publish this article.

Reviewer 3 Report

I am happy with the changes as the authors managed to accurately respond to all of my comments and recommendations.